# Ensemble Sampling

**Xiuyuan Lu**
Stanford University
lxy@stanford.edu

**Benjamin Van Roy**
Stanford University
bvr@stanford.edu

## Abstract

Thompson sampling has emerged as an effective heuristic for a broad range of online decision problems. In its basic form, the algorithm requires computing and sampling from a posterior distribution over models, which is tractable only for simple special cases. This paper develops *ensemble sampling*, which aims to approximate Thompson sampling while maintaining tractability even in the face of complex models such as neural networks. Ensemble sampling dramatically expands on the range of applications for which Thompson sampling is viable. We establish a theoretical basis that supports the approach and present computational results that offer further insight.

## 1 Introduction

Thompson sampling [8] has emerged as an effective heuristic for trading off between exploration and exploitation in a broad range of online decision problems. To select an action, the algorithm samples a model of the system from the prevailing posterior distribution and then determines which action maximizes expected immediate reward according to the sampled model. In its basic form, the algorithm requires computing and sampling from a posterior distribution over models, which is tractable only for simple special cases.

With complex models such as neural networks, exact computation of posterior distributions becomes intractable. One can resort to to the Laplace approximation, as discussed, for example, in [2, 5], but this approach is suitable only when posterior distributions are unimodal, and computations become an obstacle with complex models like neural networks because compute time requirements grow quadratically with the number of parameters. An alternative is to leverage Markov chain Monte Carlo methods, but those are computationally onerous, especially when the model is complex.

A practical approximation to Thompson sampling that can address complex models and problems requiring frequent decisions should facilitate fast incremental updating. That is, the time required per time period to learn from new data and generate a new sample model should be small and should not grow with time. Such a fast incremental method that builds on the Laplace approximation concept is presented in [5]. In this paper, we study a fast incremental method that applies more broadly, without relying on unimodality. As a sanity check we offer theoretical assurances that apply to the special case of linear bandits. We also present computational results involving simple bandit problems as well as complex neural network models that demonstrate efficacy of the approach.

Our approach is inspired by [6], which applies a similar concept to the more complex context of deep reinforcement learning, but without any theoretical analysis. The essential idea is to maintain and incrementally update an ensemble of statistically plausible models, and to sample uniformly from this set in each time period as an approximation to sampling from the posterior distribution. Each model is initially sampled from the prior, and then updated in a manner that incorporates data and random perturbations that diversify the models. The intention is for the ensemble to approximate the posterior distribution and the variance among models to diminish as the posterior concentrates. We refine this methodology and bound the incremental regret relative to exact Thompson sampling for a broad class

of online decision problems. Our bound indicates that it suffices to maintain a number of models that grows only logarithmically with the horizon of the decision problem, ensuring computational tractability of the approach.

## 2    Problem formulation

We consider a broad class of online decision problems to which Thompson sampling could, in principle, be applied, though that would typically be hindered by intractable computational requirements. We will define random variables with respect to a probability space $(\Omega, \mathbb{F}, \mathbb{P})$ endowed with a filtration $(\mathbb{F}_t : t = 0, \ldots, T)$. As a convention, random variables we index by $t$ will be $\mathbb{F}_t$-measurable, and we use $\mathbb{P}_t$ and $\mathbb{E}_t$ to denote probabilities and expectations conditioned on $\mathbb{F}_t$. The decision-maker chooses actions $A_0, \ldots, A_{T-1} \in \mathcal{A}$ and observes outcomes $Y_1, \ldots, Y_T \in \mathcal{Y}$. There is a random variable $\theta$, which represents a model index. Conditioned on $(\theta, A_{t-1})$, $Y_t$ is independent of $\mathbb{F}_{t-1}$. Further, $\mathbb{P}(Y_t = y | \theta, A_{t-1})$ does not depend on $t$. This can be thought of as a Bayesian formulation, where randomness in $\theta$ reflects prior uncertainty about which model corresponds to the true nature of the system.

We assume that $\mathcal{A}$ is finite and that each action $A_t$ is chosen by a *randomized policy* $\pi = (\pi_0, \ldots, \pi_{T-1})$. Each $\pi_t$ is $\mathbb{F}_t$-measurable, and each realization is a probability mass function over actions $\mathcal{A}$; $A_t$ is sampled independently from $\pi_t$.

The agent associates a reward $R(y)$ with each outcome $y \in \mathcal{Y}$, where the reward function $R$ is fixed and known. Let $R_t = R(Y_t)$ denote the reward realized at time $t$. Let $\overline{R}_\theta(a) = \mathbb{E}[R(Y_t) | \theta, A_{t-1} = a]$. Uncertainty about $\theta$ induces uncertainty about the true optimal action, which we denote by $A^* \in \arg\max_{a \in \mathcal{A}} \overline{R}_\theta(a)$. Let $R^* = \overline{R}_\theta(A^*)$. The $T$-period *conditional regret* when the actions $(A_0, .., A_{T-1})$ are chosen according to $\pi$ is defined by

$$\text{Regret}(T, \pi, \theta) = \mathbb{E}\left[\sum_{t=1}^{T} (R^* - R_t) \,\Big|\, \theta\right], \tag{1}$$

where the expectation is taken over the randomness in actions $A_t$ and outcomes $Y_t$, conditioned on $\theta$.

We illustrate with a couple of examples that fit our formulation.

**Example 1. (linear bandit)** *Let $\theta$ be drawn from $\Re^N$ and distributed according to a $N(\mu_0, \Sigma_0)$ prior. There is a set of $K$ actions $\mathcal{A} \subseteq \Re^N$. At each time $t = 0, 1, \ldots, T-1$, an action $A_t \in \mathcal{A}$ is selected, after which a reward $R_{t+1} = Y_{t+1} = \theta^\top A_t + W_{t+1}$ is observed, where $W_{t+1} \sim N(0, \sigma_w^2)$.*

**Example 2. (neural network)** *Let $g_\theta : \Re^N \mapsto \Re^K$ denote a mapping induced by a neural network with weights $\theta$. Suppose there are $K$ actions $\mathcal{A} \subseteq \Re^N$, which serve as inputs to the neural network, and the goal is to select inputs that yield desirable outputs. At each time $t = 0, 1, \ldots, T-1$, an action $A_t \in \mathcal{A}$ is selected, after which $Y_{t+1} = g_\theta(A_t) + W_{t+1}$ is observed, where $W_{t+1} \sim N(0, \sigma_w^2 I)$. A reward $R_{t+1} = R(Y_{t+1})$ is associated with each observation. Let $\theta$ be distributed according to a $N(\mu_0, \Sigma_0)$ prior. The idea here is that data pairs $(A_t, Y_{t+1})$ can be used to fit a neural network model, while actions are selected to trade off between generating data pairs that reduce uncertainty in neural network weights and those that offer desirable immediate outcomes.*

## 3    Algorithms

Thompson sampling offers a heuristic policy for selecting actions. In each time period, the algorithm samples an action from the posterior distribution $p_t(a) = \mathbb{P}_t(A^* = a)$ of the optimal action. In other words, Thompson sampling uses a policy $\pi_t = p_t$. It is easy to see that this is equivalent to sampling a model index $\hat{\theta}_t$ from the posterior distribution of models and then selecting an action $A_t = \arg\max_{a \in \mathcal{A}} \overline{R}_{\hat{\theta}_t}(a)$ that optimizes the sampled model.

Thompson sampling is computationally tractable for some problem classes, like the linear bandit problem, where the posterior distribution is Gaussian with parameters $(\mu_t, \Sigma_t)$ that can be updated incrementally and efficiently via Kalman filtering as outcomes are observed. However, when dealing with complex models, like neural networks, computing the posterior distribution becomes intractable. Ensemble sampling serves as an approximation to Thompson sampling for such contexts.

---
**Algorithm 1** EnsembleSampling
---
1: **Sample**: $\tilde{\theta}_{0,1}, \ldots, \tilde{\theta}_{0,M} \sim p_0$
2: **for** $t = 0, \ldots, T - 1$ **do**
3:      **Sample**: $m \sim \text{unif}(\{1, \ldots, M\})$
4:      **Act**: $A_t = \arg\max_{a \in \mathcal{A}} \overline{R}_{\tilde{\theta}_{t,m}}(a)$
5:      **Observe**: $Y_{t+1}$
6:      **Update**: $\tilde{\theta}_{t+1,1}, \ldots, \tilde{\theta}_{t+1,M}$
7: **end for**
---

The posterior can be interpreted as a distribution of "statistically plausible" models, by which we mean models that are sufficiently consistent with prior beliefs and the history of observations. With this interpretation in mind, Thompson sampling can be thought of as randomly drawing from the range of statistically plausible models. Ensemble sampling aims to maintain, incrementally update, and sample from a finite set of such models. In the spirit of particle filtering, this set of models approximates the posterior distribution. The workings of ensemble sampling are in some ways more intricate than conventional uses of particle filtering, however, because interactions between the ensemble of models and selected actions can skew the distribution.

While elements of ensemble sampling require customization, a general template is presented as Algorithm 1. The algorithm begins by sampling $M$ models from the prior distribution. Then, over each time period, a model is sampled uniformly from the ensemble, an action is selected to maximize expected reward under the sampled model, the resulting outcome is observed, and each of the $M$ models is updated. To produce an explicit algorithm, we must specify a model class, prior distribution, and algorithms for sampling from the prior and updating models.

For a concrete illustration, let us consider the linear bandit (Example 1). Though ensemble sampling is unwarranted in this case, since Thompson sampling is efficient, the linear bandit serves as a useful context for understanding the approach. Standard algorithms can be used to sample models from the $N(\mu_0, \Sigma_0)$ prior. One possible procedure for updating models maintains a covariance matrix, updating it according to

$$\Sigma_{t+1} = \left(\Sigma_t^{-1} + A_t A_t^\top / \sigma_w^2\right)^{-1},$$

and generates model parameters incrementally according to

$$\tilde{\theta}_{t+1,m} = \Sigma_{t+1}\left(\Sigma_t^{-1}\tilde{\theta}_{t,m} + A_t(R_{t+1} + \tilde{W}_{t+1,m})/\sigma_w^2\right),$$

for $m = 1, \ldots, M$, where $(\tilde{W}_{t,m} : t = 1, \ldots, T, m = 1, \ldots, M)$ are independent $N(0, \sigma_w^2)$ random samples drawn by the updating algorithm. It is easy to show that the resulting parameter vectors satisfy

$$\tilde{\theta}_{t,m} = \arg\min_\nu \left(\frac{1}{\sigma_w^2}\sum_{\tau=0}^{t-1}(R_{\tau+1} + \tilde{W}_{\tau+1,m} - A_\tau^\top \nu)^2 + (\nu - \tilde{\theta}_{0,m})^\top \Sigma_0^{-1}(\nu - \tilde{\theta}_{0,m})\right),$$

which admits an intuitive interpretation: each $\tilde{\theta}_{t,m}$ is a model fit to a randomly perturbed prior and randomly perturbed observations. As we establish in the appendix, for any deterministic sequence $A_0, \ldots, A_{t-1}$, conditioned on $\mathbb{F}_t$, the models $\tilde{\theta}_{t,1}, \ldots, \tilde{\theta}_{t,M}$ are independent and identically distributed according to the posterior distribution of $\theta$. In this sense, the ensemble approximates the posterior. It is not a new observation that, for deterministic action sequences, such a scheme generates exact samples of the posterior distribution (see, e.g., [7]). However, for stochastic action sequences selected by Algorithm 1, it is not immediately clear how well the ensemble approximates the posterior distribution. We will provide a bound in the next section which establishes that, as the number of models $M$ increases, the regret of ensemble sampling quickly approaches that of Thompson sampling.

The ensemble sampling algorithm we have described for the linear bandit problem motivates an analogous approach for the neural network model of Example 2. This approach would again begin with $M$ models, with connection weights $\tilde{\theta}_{0,1}, \ldots, \tilde{\theta}_{0,M}$ sampled from a $N(\mu_0, \Sigma_0)$ prior. It could be

natural here to let $\mu_0 = 0$ and $\Sigma_0 = \sigma_0^2 I$ for some variance $\sigma_0^2$ chosen so that the range of probable models spans plausible outcomes. To incrementally update parameters, at each time $t$, each model $m$ applies some number of stochastic gradient descent iterations to reduce a loss function of the form

$$\mathcal{L}_t(\nu) = \frac{1}{\sigma_w^2} \sum_{\tau=0}^{t-1} (Y_{\tau+1} + \tilde{W}_{\tau+1,m} - g_\nu(A_\tau))^2 + (\nu - \tilde{\theta}_{0,m})^\top \Sigma_0^{-1} (\nu - \tilde{\theta}_{0,m}).$$

We present computational results in Section 5.2 that demonstrate viability of this approach.

## 4    Analysis of ensemble sampling for the linear bandit

Past analyses of Thompson sampling have relied on independence between models sampled over time periods. Ensemble sampling introduces dependencies that may adversely impact performance. It is not immediately clear whether the degree of degradation should be tolerable and how that depends on the number of models in the ensemble. In this section, we establish a bound for the linear bandit context. Our result serves as a sanity check for ensemble sampling and offers insight that should extend to broader model classes, though we leave formal analysis beyond the linear bandit for future work.

Consider the linear bandit problem described in Example 1. Let $\pi^{\mathrm{TS}}$ and $\pi^{\mathrm{ES}}$ denote the Thompson and ensemble sampling policies for this problem, with the latter based on an ensemble of $M$ models, generated and updated according to the procedure described in Section 3. Let $R_* = \min_{a \in \mathcal{A}} \theta^\top a$ denote the worst mean reward and let $\Delta(\theta) = R^* - R_*$ denote the gap between maximal and minimal mean rewards. The following result bounds the difference in regret as a function of the gap, ensemble size, and number of actions.

**Theorem 3.** *For all $\epsilon > 0$, if*

$$M \geq \frac{4|\mathcal{A}|}{\epsilon^2} \log \frac{4|\mathcal{A}|T}{\epsilon^3},$$

*then*

$$\mathrm{Regret}(T, \pi^{\mathrm{ES}}, \theta) \leq \mathrm{Regret}(T, \pi^{\mathrm{TS}}, \theta) + \epsilon \Delta(\theta) T.$$

This inequality bounds the regret realized by ensemble sampling by a sum of the regret realized by Thompson sampling and an error term $\epsilon \Delta(\theta) T$. Since we are talking about cumulative regret, the error term bounds the per-period degradation relative to Thompson sampling by $\epsilon \Delta(\theta)$. The value of $\epsilon$ can be made arbitrarily small by increasing $M$. Hence, with a sufficiently large ensemble, the per-period loss will be small. This supports the viability of ensemble sampling.

An important implication of this result is that it suffices for the ensemble size to grow logarithmically in the horizon $T$. Since Thompson sampling requires independence between models sampled over time, in a sense, it relies on $T$ models – one per time period. So to be useful, ensemble sampling should operate effectively with a much smaller number, and the logarithmic dependence is suitable. The bound also grows with $|\mathcal{A}| \log |\mathcal{A}|$, which is manageable when there are a modest number of actions. We conjecture that a similar bound holds that depends instead on a multiple of $N \log N$, where $N$ is the linear dimension, which would offer a stronger guarantee when the number of actions becomes large or infinite, though we leave proof of this alternative bound for future work.

The bound of Theorem 3 is on a notion of regret conditioned on the realization of $\theta$. A Bayesian regret bound that removes dependence on this realization can be obtained by taking an expectation, integrating over $\theta$:

$$\mathbb{E}\left[\mathrm{Regret}(T, \pi^{\mathrm{ES}}, \theta)\right] \leq \mathbb{E}\left[\mathrm{Regret}(T, \pi^{\mathrm{TS}}, \theta)\right] + \epsilon \mathbb{E}\left[\Delta(\theta)\right] T.$$

We provide a complete proof of Theorem 3 in the appendix. Due to space constraints, we only offer a sketch here.

*Sketch of Proof.* Let $A$ denote an $\mathbb{F}_t$-adapted action process $(A_0, \ldots, A_{T-1})$. Our procedure for generating and updating models with ensemble sampling is designed so that, for any deterministic $A$, conditioned on the history of rewards $(R_1, \ldots, R_t)$, models $\tilde{\theta}_{t,1}, \ldots, \tilde{\theta}_{t,M}$ that comprise the ensemble are independent and identically distributed according to the posterior distribution of $\theta$. This can be verified via some algebra, as is done in the appendix.

Recall that $p_t(a)$ denotes the posterior probability $\mathbb{P}_t(A^* = a) = \mathbb{P}(A^* = a | A_0, R_1, \ldots, A_{t-1}, R_t)$. To explicitly indicate dependence on the action process, we will use a superscript: $p_t(a) = p_t^A(a)$. Let $\hat{p}_t^A$ denote an approximation to $p_t^A$, given by $\hat{p}_t^A(a) = \frac{1}{M} \sum_{m=1}^M \mathbb{I}\left(a = \arg\max_{a'} \tilde{\theta}_{t,m}^\top a'\right)$. Note that given an action process $A$, at time $t$ Thompson sampling would sample the next action from $p_t^A$, while ensemble sampling would sample the next action from $\hat{p}_t^A$. If $A$ is deterministic then, since $\tilde{\theta}_{t,1}, \ldots, \tilde{\theta}_{t,M}$, conditioned on the history of rewards, are i.i.d. and distributed as $\theta$, $\hat{p}_t^A$ represents an empirical distribution of samples drawn from $p_t^A$. It follows from this and Sanov's Theorem that, for any deterministic $A$,

$$\mathbb{P}\left(d_{KL}(\hat{p}_t^A \| p_t^A) \geq \epsilon | \theta\right) \leq (M+1)^{|\mathcal{A}|} e^{-M\epsilon}.$$

A naive application of the union bound over all deterministic action sequences would establish that, for any $A$ (deterministic or stochastic),

$$\mathbb{P}\left(d_{KL}(\hat{p}_t^A \| p_t^A) \geq \epsilon | \theta\right) \leq \mathbb{P}\left(\max_{a \in \mathcal{A}^t} d_{KL}(\hat{p}_t^a \| p_t^a) \geq \epsilon \Big| \theta\right) \leq |\mathcal{A}|^t (M+1)^{|\mathcal{A}|} e^{-M\epsilon}$$

However, our proof takes advantage of the fact that, for any deterministic $A$, $p_t^A$ and $\hat{p}_t^A$ do not depend on the ordering of past actions and observations. To make it precise, we encode the sequence of actions in terms of action counts $c_0, \ldots, c_{T-1}$. In particular, let $c_{t,a} = |\{\tau \leq t : A_\tau = a\}|$ be the number of times that action $a$ has been selected by time $t$. We apply a coupling argument that introduces dependencies between the noise terms $W_t$ and action counts, without changing the distributions of any observable variables. We let $(Z_{n,a} : n \in \mathbb{N}, a \in \mathcal{A})$ be i.i.d. $N(0,1)$ random variables, and let $W_{t+1} = Z_{c_{t,A_t}, A_t}$. Similarly, we let $(\tilde{Z}_{n,a,m} : n \in \mathbb{N}, a \in \mathcal{A}, m = 1, \ldots, M)$ be i.i.d $N(0,1)$ random variables, and let $\tilde{W}_{t+1,m} = \tilde{Z}_{c_{t,A_t}, A_t, m}$. To make explicit the dependence on $A$, we will use a superscript and write $c_t^A$ to denote the action counts at time $t$ when the action process is given by $A$. It is not hard to verify, as is done in the appendix, that if $a, \bar{a} \in \mathcal{A}^T$ are two deterministic action sequences such that $c_{t-1}^a = c_{t-1}^{\bar{a}}$, then $p_t^a = p_t^{\bar{a}}$ and $\hat{p}_t^a = \hat{p}_t^{\bar{a}}$. This allows us to apply the union bound over action counts, instead of action sequences, and we get that for any $A$ (deterministic or stochastic),

$$\mathbb{P}\left(d_{KL}(\hat{p}_t^A \| p_t^A) \geq \epsilon | \theta\right) \leq \mathbb{P}\left(\max_{c_{t-1}^a : a \in \mathcal{A}^t} d_{KL}(\hat{p}_t^a \| p_t^a) \geq \epsilon \Big| \theta\right) \leq (t+1)^{|\mathcal{A}|} (M+1)^{|\mathcal{A}|} e^{-M\epsilon}.$$

Now, we specialize the action process $A$ to the action sequence $A_t = A_t^{\text{ES}}$ selected by ensemble sampling, and we will omit the superscripts in $p_t^A$ and $\hat{p}_t^A$. We can decompose the per-period regret of ensemble sampling as

$$\mathbb{E}\left[R^* - \theta^\top A_t | \theta\right] = \mathbb{E}\left[(R^* - \theta^\top A_t)\mathbb{I}\left(d_{KL}(\hat{p}_t \| p_t) \geq \epsilon\right) | \theta\right]$$
$$+ \mathbb{E}\left[(R^* - \theta^\top A_t)\mathbb{I}\left(d_{KL}(\hat{p}_t \| p_t) < \epsilon\right) | \theta\right]. \quad (2)$$

The first term can be bounded by

$$\mathbb{E}\left[(R^* - \theta^\top A_t)\mathbb{I}\left(d_{KL}(\hat{p}_t \| p_t) \geq \epsilon\right) | \theta\right] \leq \Delta(\theta)\mathbb{P}\left(d_{KL}(\hat{p}_t \| p_t) \geq \epsilon | \theta\right)$$
$$\leq \Delta(\theta)(t+1)^{|\mathcal{A}|}(M+1)^{|\mathcal{A}|} e^{-M\epsilon}.$$

To bound the second term, we will use another coupling argument that couples the actions that would be selected by ensemble sampling with those that would be selected by Thompson sampling. Let $A_t^{\text{TS}}$ denote the action that Thompson sampling would select at time $t$. On $\{d_{\text{KL}}(\hat{p}_t \| p_t) \leq \epsilon\}$, we have $\|\hat{p}_t - p_t\|_{\text{TV}} \leq \sqrt{2\epsilon}$ by Pinsker's inequality. Conditioning on $\hat{p}_t$ and $p_t$, if $d_{\text{KL}}(\hat{p}_t \| p_t) \leq \epsilon$, we can construct random variables $\tilde{A}_t^{\text{ES}}$ and $\tilde{A}_t^{\text{TS}}$ such that they have the same distributions as $A_t^{\text{ES}}$ and $A_t^{\text{TS}}$, respectively. Using maximal coupling, we can make $\tilde{A}_t^{\text{ES}} = \tilde{A}_t^{\text{TS}}$ with probability at least $1 - \frac{1}{2}\|\hat{p}_t - p_t\|_{\text{TV}} \geq 1 - \sqrt{\epsilon/2}$. Then, the second term of the sum in (2) can be decomposed into

$$\mathbb{E}\left[(R^* - \theta^\top A_t)\mathbb{I}\left(d_{\text{KL}}(\hat{p}_t \| p_t) \leq \epsilon\right) | \theta\right]$$
$$= \mathbb{E}\left[\mathbb{E}\left[(R^* - \theta^\top \tilde{A}_t^{\text{ES}})\mathbb{I}\left(d_{\text{KL}}(\hat{p}_t \| p_t) \leq \epsilon, \tilde{A}_t^{\text{ES}} = \tilde{A}_t^{\text{TS}}\right) | \hat{p}_t, p_t, \theta\right] | \theta\right]$$
$$+ \mathbb{E}\left[\mathbb{E}\left[(R^* - \theta^\top \tilde{A}_t^{\text{ES}})\mathbb{I}\left(d_{\text{KL}}(\hat{p}_t \| p_t) \leq \epsilon, \tilde{A}_t^{\text{ES}} \neq \tilde{A}_t^{\text{TS}}\right) | \hat{p}_t, p_t, \theta\right] | \theta\right],$$

which, after some algebraic manipulations, leads to

$$\mathbb{E}\left[(R^* - \theta^\top A_t)\mathbb{I}\left(d_{KL}(\hat{p}_t\|p_t) < \epsilon\right)|\theta\right] \leq \mathbb{E}\left[R^* - \theta^\top A_t^{\text{TS}}|\theta\right] + \sqrt{\epsilon/2}\,\Delta(\theta).$$

The result then follows from some straightforward algebra. □

## 5 Computational results

In this section, we present computational results that demonstrate viability of ensemble sampling. We will start with a simple case of independent Gaussian bandits in Section 5.1 and move on to more complex models of neural networks in Section 5.2. Section 5.1 serves as a sanity check for the empirical performance of ensemble sampling, as Thompson sampling can be efficiently applied in this case and we are able to compare the performances of these two algorithms. In addition, we provide simulation results that demonstrate how the ensemble size grows with the number of actions. Section 5.2 goes beyond our theoretical analysis in Section 4 and gives computational evidence of the efficacy of ensemble sampling when applied to more complex models such as neural networks. We show that ensemble sampling, even with a few models, achieves efficient learning and outperforms $\epsilon$-greedy and dropout on the example neural networks.

### 5.1 Gaussian bandits with independent arms

We consider a Gaussian bandit with $K$ actions, where action $k$ has mean reward $\theta_k$. Each $\theta_k$ is drawn i.i.d. from $N(0,1)$. During each time step $t = 0, \ldots, T-1$, we select an action $k \in \{1, \ldots, K\}$ and observe reward $R_{t+1} = \theta_k + W_{t+1}$, where $W_{t+1} \sim N(0,1)$. Note that this is a special case of Example 1. Since the posterior distribution of $\theta$ can be explicitly computed in this case, we use it as a sanity check for the performance of ensemble sampling.

Figure 1a shows the per-period regret of Thompson sampling and ensemble sampling applied to a Gaussian bandit with 50 independent arms. We see that as the number of models increases, ensemble sampling better approximates Thompson sampling. The results were averaged over 2,000 realizations. Figure 1b shows the minimum number of models required so that the expected per-period regret of ensemble sampling is no more than $\epsilon$ plus the expected per-period regret of Thompson sampling at some large time horizon $T$ across different numbers of actions. All results are averaged over 10,000 realizations. We chose $T = 2000$ and $\epsilon = 0.03$. The plot shows that the number of models needed seems to grow sublinearly with the number of actions, which is stronger than the bound proved in Section 4.

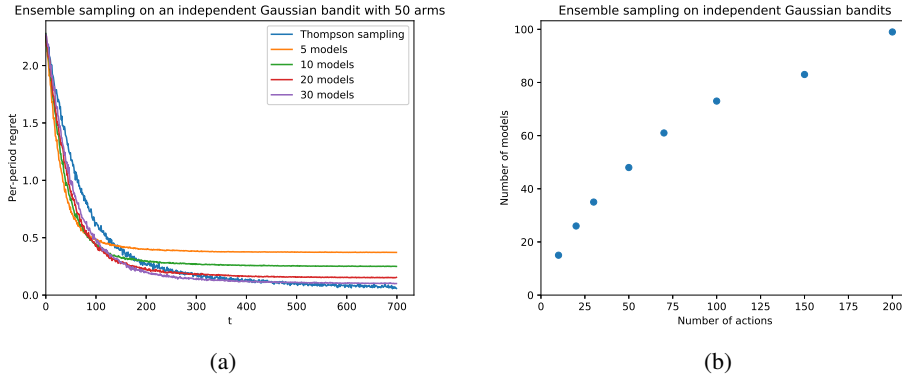

(a)                              (b)

Figure 1: (a) Ensemble sampling compared with Thompson sampling on a Gaussian bandit with 50 independent arms. (b) Minimum number of models required so that the expected per-period regret of ensemble sampling is no more than $\epsilon = 0.03$ plus the expected per-period regret of Thompson sampling at $T = 2000$ for Gaussian bandits across different numbers of arms.

### 5.2 Neural networks

In this section, we follow Example 2 and show computational results of ensemble sampling applied to neural networks. Figure 2 shows $\epsilon$-greedy and ensemble sampling applied to a bandit problem

where the mapping from actions to expected rewards is represented by a neuron. More specifically, we have a set of $K$ actions $\mathcal{A} \subseteq \Re^N$. The mean reward of selecting an action $a \in \mathcal{A}$ is given by $g_\theta(a) = \max(0, \theta^\top a)$, where weights $\theta \in \Re^N$ are drawn from $N(0, \lambda I)$. During each time period, we select an action $A_t \in \mathcal{A}$ and observe reward $R_{t+1} = g_\theta(A_t) + Z_{t+1}$, where $Z_{t+1} \sim N(0, \sigma_z^2)$. We set the input dimension $N = 100$, number of actions $K = 100$, prior variance $\lambda = 10$, and noise variance $\sigma_z^2 = 100$. Each dimension of each action was sampled uniformly from $[-1, 1]$, except for the last dimension, which was set to 1.

In Figure 3, we consider a bandit problem where the mapping from actions to expected rewards is represented by a two-layer neural network with weights $\theta \equiv (W_1, W_2)$, where $W_1 \in \Re^{D \times N}$ and $W_2 \in \Re^D$. Each entry of the weight matrices is drawn independently from $N(0, \lambda)$. There is a set of $K$ actions $\mathcal{A} \subseteq \Re^N$. The mean reward of choosing an action $a \in \mathcal{A}$ is $g_\theta(a) = W_2^\top \max(0, W_1 a)$. During each time period, we select an action $A_t \in \mathcal{A}$ and observe reward $R_{t+1} = g_\theta(A_t) + Z_{t+1}$, where $Z_{t+1} \sim N(0, \sigma_z^2)$. We used $N = 100$ for the input dimension, $D = 50$ for the dimension of the hidden layer, number of actions $K = 100$, prior variance $\lambda = 1$, and noise variance $\sigma_z^2 = 100$. Each dimension of each action was sampled uniformly from $[-1, 1]$, except for the last dimension, which was set to 1.

Ensemble sampling with $M$ models starts by sampling $\tilde{\theta}_m$ from the prior distribution independently for each model $m$. At each time step, we pick a model $m$ uniformly at random and apply the greedy action with respect to that model. We update the ensemble incrementally. During each time period, we apply a few steps of stochastic gradient descent for each model $m$ with respect to the loss function

$$\mathcal{L}_t(\theta) = \frac{1}{\sigma_z^2} \sum_{\tau=0}^{t-1} (R_{\tau+1} + \tilde{Z}_{\tau+1,m} - g_\theta(A_\tau))^2 + \frac{1}{\lambda} \|\theta - \tilde{\theta}_m\|_2^2,$$

where perturbations $(\tilde{Z}_{t,m} : t = 1, \ldots, T, m = 1, \ldots, M)$ are drawn i.i.d. from $N(0, \sigma_z^2)$.

Besides ensemble sampling, there are other heuristics for sampling from an approximate posterior distribution over neural networks, which may be used to develop approximate Thompson sampling. Gal and Ghahramani proposed an approach based on dropout [4] to approximately sample from a posterior over neural networks. In Figure 3, we include results from using dropout to approximate Thompson sampling on the two-layer neural network bandit.

To facilitate gradient flow, we used leaky ReLUs of the form $\max(0.01x, x)$ internally in all agents, while the target neural nets still use regular ReLUs as described above. We took 3 stochastic gradient steps with a minibatch size of 64 for each model update. We used a learning rate of `1e-1` for $\epsilon$-greedy and ensemble sampling, and a learning rate of `1e-2`, `1e-2`, `2e-2`, and `5e-2` for dropout with dropping probabilities 0.25, 0.5, 0.75, and 0.9 respectively. All results were averaged over around 1,000 realizations.

Figure 2 plots the per-period regret of $\epsilon$-greedy and ensemble sampling on the single neuron bandit. We see that ensemble sampling, even with 10 models, performs better than $\epsilon$-greedy with the best tuned parameters. Increasing the size of the ensemble further improves the performance. An ensemble of size 50 achieves orders of magnitude lower regret than $\epsilon$-greedy.

Figure 3a and 3b show different versions of $\epsilon$-greedy applied to the two-layer neural network model. We see that $\epsilon$-greedy with an annealing schedule tends to perform better than a fixed $\epsilon$. Figure 3c plots the per-period regret of the dropout approach with different dropping probabilities, which seems to perform worse than $\epsilon$-greedy. Figure 3d plots the per-period regret of ensemble sampling on the neural net bandit. Again, we see that ensemble sampling, with a moderate number of models, outperforms the other approaches by a significant amount.

# 6   Conclusion

Ensemble sampling offers a potentially efficient means to approximate Thompson sampling when using complex models such as neural networks. We have provided an analysis that offers theoretical assurances for the case of linear bandit models and computational results that demonstrate efficacy with complex neural network models.

We are motivated largely by the need for effective exploration methods that can efficiently be applied in conjunction with complex models such as neural networks. Ensemble sampling offers one approach

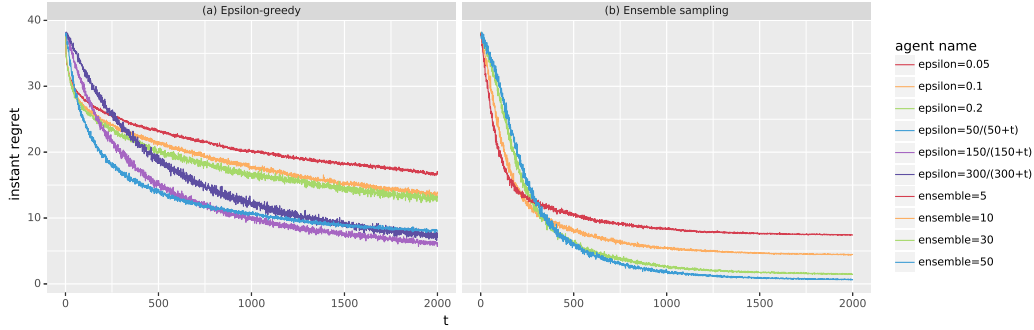

Figure 2: (a) $\epsilon$-greedy and (b) ensemble sampling applied to a single neuron bandit.

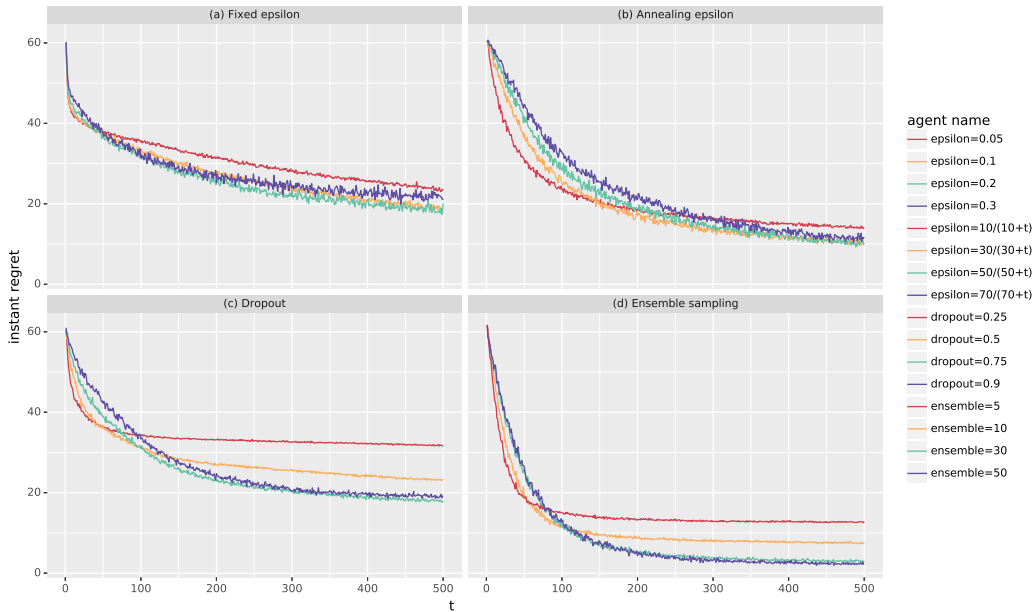

Figure 3: (a) Fixed $\epsilon$-greedy, (b) annealing $\epsilon$-greedy, (c) dropout, and (d) ensemble sampling applied to a two-layer neural network bandit.

to representing uncertainty in neural network models, and there are others that might also be brought to bear in developing approximate versions of Thompson sampling [1, 4]. The analysis of various other forms of approximate Thompson sampling remains open.

Ensemble sampling loosely relates to ensemble learning methods [3], though an important difference in motivation lies in the fact that the latter learns multiple models for the purpose of generating a more accurate model through their combination, while the former learns multiple models to reflect uncertainty in the posterior distribution over models. That said, combining the two related approaches may be fruitful. In particular, there may be practical benefit to learning many forms of models (neural networks, tree-based models, etc.) and viewing the ensemble as representing uncertainty from which one can sample.

## Acknowledgments

This work was generously supported by a research grant from Boeing and a Marketing Research Award from Adobe.

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
