[Supplementary Material]

# A  Proof of Theorem 3

Without loss of generality, we assume that $\sigma_w^2 = 1$. Recall from Section 3 the procedure for generating and updating models with ensemble sampling. First, $\tilde{\theta}_{0,1}, \ldots, \tilde{\theta}_{0,M}$ are sampled i.i.d. from $N(\mu_0, \Sigma_0)$. Then, these vectors are adapted according to

$$\tilde{\theta}_{t,m} = \arg\min_{\nu} \left( \sum_{\tau=0}^{t-1} (R_{\tau+1} + \tilde{W}_{\tau+1,m} - A_\tau^\top \nu)^2 + (\nu - \tilde{\theta}_{0,m})^\top \Sigma_0^{-1} (\nu - \tilde{\theta}_{0,m}) \right).$$

Note that we have not yet specified how actions are selected. In the formulation we have put forth, each $A_t$ could be any $\mathbb{F}_t$-measureable random variable. We will denote by $A$ the $\mathbb{F}_t$-adapted process $(A_0, \ldots, A_{T-1})$. We say $A$ is deterministic if there exist $a_0, \ldots, a_{T-1} \in \mathcal{A}$ such that $A_t = a_t$ for $t = 0, \ldots, T-1$ with probability one.

**Lemma 1.** *If $A$ is deterministic, then, conditioned on $(R_1, \ldots, R_t)$, $\tilde{\theta}_{t,1}, \cdots, \tilde{\theta}_{t,M}$ are i.i.d. $N(\mu_t, \Sigma_t)$ random variables, where $\mu_t = \mathbb{E}[\theta|\mathbb{F}_t]$ and $\Sigma_t = \mathbb{E}[(\theta - \mu_t)(\theta - \mu_t)^\top | \mathbb{F}_t]$.*

*Proof.* Say $A = (a_0, \ldots, a_{T-1})$, where $a_0, \ldots, a_{T-1} \in \mathcal{A}$. Let $X$ be an $t \times N$ matrix with the $j^{\text{th}}$ row equal to $a_{j-1}$. Let $y = (R_1, \ldots, R_t)^\top$. Then,

$$\mu_t = \arg\min_{\nu} \left( \sum_{\tau=0}^{t-1} (R_{\tau+1} - a_\tau^\top \nu)^2 + (\nu - \mu_0)^\top \Sigma_0^{-1} (\nu - \mu_0) \right)$$

$$= \left( X^\top X + \Sigma_0^{-1} \right)^{-1} \left( X^\top y + \Sigma_0^{-1} \mu_0 \right),$$

and

$$\Sigma_t = \left( X^\top X + \Sigma_0^{-1} \right)^{-1}.$$

For any $m = 1, \ldots, M$, we have

$$\tilde{\theta}_{t,m} = \arg\min_{\nu} \left( \sum_{\tau=0}^{t-1} (R_{\tau+1} + \tilde{W}_{\tau+1,m} - a_\tau^\top \nu)^2 + (\nu - \tilde{\theta}_{0,m})^\top \Sigma_0^{-1} (\nu - \tilde{\theta}_{0,m}) \right)$$

$$= \left( X^\top X + \Sigma_0^{-1} \right)^{-1} \left( X^\top (y + \tilde{W}_m) + \Sigma_0^{-1} \tilde{\theta}_{0,m} \right),$$

where $\tilde{W}_m = (\tilde{W}_{1,m}, \ldots, \tilde{W}_{t,m})^\top$. We first observe that, conditioned on $y$, $\tilde{\theta}_{t,m}$ follows a normal distribution, since it is affine in $\tilde{\theta}_{0,m}$ and $\tilde{W}_m$. Next, we check its mean and covariance. Since $\tilde{W}_m$ and $\tilde{\theta}_{0,m}$ are independently sampled, we have

$$\mathbb{E}\left[ \tilde{\theta}_{t,m} | y \right] = \left( X^\top X + \Sigma_0^{-1} \right)^{-1} \left( X^\top \left( y + \mathbb{E}\left[ \tilde{W}_m | y \right] \right) + \Sigma_0^{-1} \mathbb{E}\left[ \tilde{\theta}_{0,m} | y \right] \right) = \mu_t,$$

and

$$\text{Cov}\left[ \tilde{\theta}_{t,m} | y \right] = \left( X^\top X + \Sigma_0^{-1} \right)^{-1} \left( X^\top \mathbb{E}\left[ \tilde{W}_m \tilde{W}_m^\top | y \right] X \right.$$
$$\left. + \Sigma_0^{-1} \mathbb{E}\left[ (\tilde{\theta}_{0,m} - \mu_0)(\tilde{\theta}_{0,m} - \mu_0)^\top | y \right] \Sigma_0^{-1} \right) \left( X^\top X + \Sigma_0^{-1} \right)^{-1} = \Sigma_t.$$

Therefore, if $A$ is deterministic, then for each $m = 1, \ldots, M$, $\tilde{\theta}_{t,m}$ is a $N(\mu_t, \Sigma_t)$ random variable conditioned on $(R_1, \ldots, R_t)$. Further, since $\tilde{W}_m$ and $\tilde{\theta}_{0,m}$, $m = 1, \ldots, M$ are all independent, $\tilde{\theta}_{t,1}, \ldots, \tilde{\theta}_{t,M}$ are independent. $\qquad\square$

Recall that $p_t(a)$ denotes the posterior probability $\mathbb{P}_t(A^* = a) = \mathbb{P}(A^* = a | A_0, R_1, \ldots, A_{t-1}, R_t)$. To explicitly indicate dependence on the action process, we will use a superscript: $p_t(a) = p_t^A(a)$. Let $\hat{p}_t^A$ denote an approximation to $p_t^A$, given by $\hat{p}_t^A(a) = \frac{1}{M} \sum_{m=1}^{M} \mathbb{I}\left( a = \arg\max_{a'} \tilde{\theta}_{t,m}^\top a' \right)$. Note that given an action process $A$, at time $t$ Thompson sampling would sample the next action from $p_t^A$, while ensemble sampling would sample the next action from $\hat{p}_t^A$.

The following lemma shows that for any deterministic action sequence, conditioned on $\theta$, the action distribution that ensemble sampling would sample from is close to the action distribution that Thompson sampling would sample from with high probability.

**Lemma 2.** *For any deterministic action sequence $a \in \mathcal{A}^T$,*

$$\mathbb{P}\left(d_{KL}(\hat{p}_t^a \| p_t^a) \geq \epsilon \,|\, \theta\right) \leq (M+1)^{|\mathcal{A}|} e^{-M\epsilon}.$$

*Proof.* If $a \in \mathcal{A}^T$ is deterministic, then conditioned on $(R_1, \ldots, R_t)$, $\hat{p}_t^a$ and $p_t^a$ are independent of $\theta$. Thus, we have

$$\mathbb{P}\left(d_{\mathrm{KL}}(\hat{p}_t^a \| p_t^a) \geq \epsilon \,|\, \theta\right) = \mathbb{E}\left[\mathbb{E}\left[\mathbb{I}\left(d_{\mathrm{KL}}(\hat{p}_t^a \| p_t^a) \geq \epsilon\right) | R_1, \ldots, R_t\right] | \theta\right].$$

By Lemma 1, $\tilde{\theta}_{t,1}, \ldots, \tilde{\theta}_{t,M}$, conditioned on $(R_1, \ldots, R_t)$, are i.i.d. and distributed as the posterior of $\theta$. Thus, $\hat{p}_t^a$ represents an empirical distribution of samples drawn from $p_t^a$. Sanov's Theorem then implies that

$$\mathbb{P}\left(d_{\mathrm{KL}}(\hat{p}_t^a \| p_t^a) \geq \epsilon | R_1, \ldots, R_t\right) \leq (M+1)^{|\mathcal{A}|} e^{-M\epsilon}.$$

The result follows. □

Next, we will establish results for any $\mathbb{F}_t$-adapted action process $A$, deterministic or stochastic. To do so, it is useful to introduce the notion of action counts. One way of encoding the sequence of actions is in terms of counts $c_0, \ldots, c_{T-1}$. In particular, let $c_{t,a} = |\{\tau \leq t : A_\tau = a\}|$ be the number of times that action $a$ has been selected by time $t$. Each $c_t$ takes values in a set

$$C_t = \left\{ \overline{c} \in \mathbb{N}^{|\mathcal{A}|} : \sum_{a \in \mathcal{A}} \overline{c}_a = t+1 \right\}.$$

Since $c_t$ has $|\mathcal{A}|$ components, and each component takes a value in $\{0, \ldots, t+1\}$, we have

$$|C_t| \leq (t+2)^{|\mathcal{A}|}.$$

Sometimes, we use a superscript and write $c_t^A$ to explicitly denote the dependence on action process $A$.

We now introduce dependencies between the noise terms $W_t$ and action counts, without changing the distributions of any observable variables. This will turn out to be useful when we take the union bound later. We let $(Z_{n,a} : n \in \mathbb{N}, a \in \mathcal{A})$ be i.i.d. $N(0,1)$ random variables, and let $W_{t+1} = Z_{c_{t,A_t},A_t}$. Similarly, we let $(\tilde{Z}_{n,a,m} : n \in \mathbb{N}, a \in \mathcal{A}, m = 1, \ldots, M)$ be i.i.d $N(0,1)$ random variables, and let $\tilde{W}_{t+1,m} = \tilde{Z}_{c_{t,A_t},A_t,m}$.

The following lemma establishes that, for any deterministic action sequence $a \in \mathcal{A}^T$, $p_t^a$ and $\hat{p}_t^a$ depend on $a$ only through its action counts, $c_{t-1}^a$; in other words, $p_t^a$ and $\hat{p}_t^a$ do not depend on the ordering of past actions and observations.

**Lemma 3.** *For any $t = 0, \ldots, T-1$, if $a, \overline{a} \in \mathcal{A}^T$ are deterministic sequences such that $c_{t-1}^a = c_{t-1}^{\overline{a}}$, then $p_t^a = p_t^{\overline{a}}$ and $\hat{p}_t^a = \hat{p}_t^{\overline{a}}$.*

*Proof.* Recall that $R_{\tau+1} = \theta^\top A_\tau + W_{\tau+1}$, where $W_{\tau+1} = Z_{c_{\tau,A_\tau},A_\tau}$. This means that we observe the same reward the first time we take an action, regardless of where that action appears in the action sequence. Similarly, for all action sequences, we observe the same reward the second time we take that action, and so on. Therefore, if $c_{t-1}^a = c_{t-1}^{\overline{a}}$, we have $\mu_t^a = \mu_t^{\overline{a}}$ and $\Sigma_t^a = \Sigma_t^{\overline{a}}$, which implies that $p_t^a = p_t^{\overline{a}}$. By the same reasoning, since $\tilde{W}_{\tau+1,m} = \tilde{Z}_{c_{\tau,A_\tau},A_\tau,m}$ for all $\tau$ and $m$, both action sequences would yield the same model parameters, and it follows that $\hat{p}_t^a = \hat{p}_t^{\overline{a}}$. □

**Lemma 4.** *For any $\mathbb{F}_t$-adapted process $A$,*

$$\mathbb{P}\left(d_{KL}(\hat{p}_t^A \| p_t^A) > \epsilon \,|\, \theta\right) \leq (t+1)^{|\mathcal{A}|}(M+1)^{|\mathcal{A}|} e^{-M\epsilon}.$$

*Proof.* We have

$$
\begin{aligned}
\mathbb{P}\left(d_{\mathrm{KL}}(\hat{p}_t^A \| p_t^A) > \epsilon \,|\, \theta\right) \quad &\leq \quad \mathbb{P}\left(\max_{a \in \mathcal{A}^T} d_{\mathrm{KL}}(\hat{p}_t^a \| p_t^a) > \epsilon \,\Big|\, \theta\right) \\
&\overset{(a)}{=} \quad \mathbb{P}\left(\max_{c_{t-1}^a : a \in \mathcal{A}^T} d_{\mathrm{KL}}(\hat{p}_t^a \| p_t^a) > \epsilon \,\Big|\, \theta\right) \\
&\overset{(b)}{\leq} \quad \sum_{c_{t-1}^a : a \in \mathcal{A}^T} \mathbb{P}\left(d_{\mathrm{KL}}(\hat{p}_t^a \| p_t^a) > \epsilon \,|\, \theta\right) \\
&\overset{(c)}{\leq} \quad (t+1)^{|\mathcal{A}|}(M+1)^{|\mathcal{A}|} e^{-M\epsilon},
\end{aligned}
$$

where $(a)$ follows from Lemma 3, $(b)$ follows from the union bound, and $(c)$ follows from Lemma 2 and the fact that the total number of counts $|C_{t-1}| \leq (t+1)^{|\mathcal{A}|}$. $\qquad\square$

Now, we specialize the action process $A$ to the action sequence $A_t = A_t^{\mathrm{ES}}$ selected by ensemble sampling, and we will omit the superscripts in $p_t^A$ and $\hat{p}_t^A$.

The expected cumulative regret of ensemble sampling conditioned on $\theta$ can be decomposed as

$$
\begin{aligned}
\mathrm{Regret}(T, \pi^{\mathrm{ES}}, \theta) &= \sum_{t=0}^{T-1} \mathbb{E}\left[R^* - \theta^\top A_t^{\mathrm{ES}} \,|\, \theta\right] \\
&= \sum_{t=0}^{T-1} \Big( \mathbb{E}\left[(R^* - \theta^\top A_t^{\mathrm{ES}}) \mathbb{I}\left(d_{\mathrm{KL}}(\hat{p}_t \| p_t) > \epsilon\right) |\theta\right] \\
&\qquad\qquad + \mathbb{E}\left[(R^* - \theta^\top A_t^{\mathrm{ES}}) \mathbb{I}\left(d_{\mathrm{KL}}(\hat{p}_t \| p_t) \leq \epsilon\right) |\theta\right] \Big).
\end{aligned}
$$

We will bound the per-period regret for the case where the divergence $d_{\mathrm{KL}}(\hat{p}_t \| p_t)$ is large and the case where the divergence is small, respectively.

**Lemma 5.** *For any $t = 0, \dots, T-1$,*
$$
\mathbb{E}\left[(R^* - \theta^\top A_t^{\mathrm{ES}}) \mathbb{I}\left(d_{KL}(\hat{p}_t \| p_t) > \epsilon\right) |\theta\right] \leq (t+1)^{|\mathcal{A}|}(M+1)^{|\mathcal{A}|} e^{-M\epsilon} \Delta(\theta).
$$

*Proof.* This follows directly from the definition of $\Delta(\theta)$ and Lemma 4. $\qquad\square$

**Assumption 6.** *For simplicity, assume $0 < \epsilon < 1$ and $0 < \delta < 1$ are such that $\frac{|\mathcal{A}|T}{\epsilon\delta} \geq 9$.*

**Lemma 7.** *If the size of the ensemble satisfies*
$$
M \geq \frac{2|\mathcal{A}|}{\epsilon} \log \frac{|\mathcal{A}|T}{\epsilon\delta},
$$
*then*
$$
\sum_{t=0}^{T-1} \mathbb{E}\left[(R^* - \theta^\top A_t^{\mathrm{ES}}) \mathbb{I}\left(d_{KL}(\hat{p}_t \| p_t) > \epsilon\right) |\theta\right] \leq \delta \Delta(\theta) T.
$$

*Proof.* We show that if $M$ satisfies the condition above, then
$$
T^{|\mathcal{A}|}(M+1)^{|\mathcal{A}|} e^{-M\epsilon} \leq \delta,
$$
or, equivalently,
$$
M - \frac{|\mathcal{A}|}{\epsilon} \log(M+1) \geq \frac{1}{\epsilon}\left(|\mathcal{A}| \log T + \log \frac{1}{\delta}\right).
$$
We have
$$
\begin{aligned}
&M - \frac{|\mathcal{A}|}{\epsilon} \log(M+1) - \frac{1}{\epsilon}\left(|\mathcal{A}| \log T + \log \frac{1}{\delta}\right) \\
&\geq \quad \frac{2|\mathcal{A}|}{\epsilon} \log \frac{|\mathcal{A}|T}{\epsilon\delta} - \frac{|\mathcal{A}|}{\epsilon} \log \left(\frac{4|\mathcal{A}|}{\epsilon} \log \frac{|\mathcal{A}|T}{\epsilon\delta}\right) - \frac{|\mathcal{A}|}{\epsilon} \log \frac{T}{\delta} \\
&= \quad \frac{|\mathcal{A}|}{\epsilon} \log \left(\frac{|\mathcal{A}|}{\epsilon} \cdot \frac{|\mathcal{A}|T}{\epsilon\delta}\right) - \frac{|\mathcal{A}|}{\epsilon} \log \left(\frac{4|\mathcal{A}|}{\epsilon} \log \frac{|\mathcal{A}|T}{\epsilon\delta}\right) \\
&\geq \quad 0,
\end{aligned}
$$

since Assumption 6 implies that $\frac{|\mathcal{A}|T}{\epsilon\delta} \geq 4\log\frac{|\mathcal{A}|T}{\epsilon\delta}$. The result then follows from Lemma 5. $\qquad\square$

**Lemma 8.** *Let $\pi^{\mathrm{TS}}$ denote the Thompson sampling policy. We have*

$$\sum_{t=0}^{T-1}\mathbb{E}\left[(R^* - \theta^\top A_t^{\mathrm{ES}})\mathbb{I}\left(d_{KL}(\hat{p}_t\|p_t) \leq \epsilon\right)|\theta\right] \leq \mathrm{Regret}(T, \pi^{\mathrm{TS}}, \theta) + \sqrt{\epsilon/2}\,\Delta(\theta)T.$$

*Proof.* We apply a coupling argument that couples the actions that would be selected by ensemble sampling with those that would be selected by Thompson sampling. Let $A_t^{\mathrm{TS}}$ denote the action that Thompson sampling would select at time $t$. On $\{d_{\mathrm{KL}}(\hat{p}_t\|p_t) \leq \epsilon\}$, Pinsker's inequality implies that

$$\|\hat{p}_t - p_t\|_{\mathrm{TV}} \leq \sqrt{2\epsilon}.$$

Conditioning on $\hat{p}_t$ and $p_t$, if $d_{\mathrm{KL}}(\hat{p}_t\|p_t) \leq \epsilon$, we construct random variables $\tilde{A}_t^{\mathrm{ES}}$ and $\tilde{A}_t^{\mathrm{TS}}$ such that they have the same distribution as $A_t^{\mathrm{ES}}$ and $A_t^{\mathrm{TS}}$, respectively. Using maximal coupling, we can make $\tilde{A}_t^{\mathrm{ES}} = \tilde{A}_t^{\mathrm{TS}}$ with probability at least $1 - \frac{1}{2}\|\hat{p}_t - p_t\|_{\mathrm{TV}} \geq 1 - \sqrt{\epsilon/2}$. Then,

$$\mathbb{E}\left[(R^* - \theta^\top A_t^{\mathrm{ES}})\mathbb{I}\left(d_{\mathrm{KL}}(\hat{p}_t\|p_t) \leq \epsilon\right)|\theta\right]$$
$$= \mathbb{E}\left[\mathbb{E}\left[(R^* - \theta^\top A_t^{\mathrm{ES}})\mathbb{I}\left(d_{\mathrm{KL}}(\hat{p}_t\|p_t) \leq \epsilon\right)|\hat{p}_t, p_t, \theta\right]|\theta\right]$$
$$= \mathbb{E}\left[\mathbb{E}\left[(R^* - \theta^\top \tilde{A}_t^{\mathrm{ES}})\mathbb{I}\left(d_{\mathrm{KL}}(\hat{p}_t\|p_t) \leq \epsilon\right)|\hat{p}_t, p_t, \theta\right]|\theta\right]$$
$$= \mathbb{E}\left[\mathbb{E}\left[(R^* - \theta^\top \tilde{A}_t^{\mathrm{ES}})\mathbb{I}\left(d_{\mathrm{KL}}(\hat{p}_t\|p_t) \leq \epsilon, \tilde{A}_t^{\mathrm{ES}} = \tilde{A}_t^{\mathrm{TS}}\right)|\hat{p}_t, p_t, \theta\right]|\theta\right]$$
$$+ \mathbb{E}\left[\mathbb{E}\left[(R^* - \theta^\top \tilde{A}_t^{\mathrm{ES}})\mathbb{I}\left(d_{\mathrm{KL}}(\hat{p}_t\|p_t) \leq \epsilon, \tilde{A}_t^{\mathrm{ES}} \neq \tilde{A}_t^{\mathrm{TS}}\right)|\hat{p}_t, p_t, \theta\right]|\theta\right]$$

On the first part of the sum, we have

$$\mathbb{E}\left[\mathbb{E}\left[(R^* - \theta^\top \tilde{A}_t^{\mathrm{ES}})\mathbb{I}\left(d_{\mathrm{KL}}(\hat{p}_t\|p_t) \leq \epsilon, \tilde{A}_t^{\mathrm{ES}} = \tilde{A}_t^{\mathrm{TS}}\right)|\hat{p}_t, p_t, \theta\right]|\theta\right]$$
$$= \mathbb{E}\left[\mathbb{E}\left[(R^* - \theta^\top \tilde{A}_t^{\mathrm{TS}})\mathbb{I}\left(d_{\mathrm{KL}}(\hat{p}_t\|p_t) \leq \epsilon, \tilde{A}_t^{\mathrm{ES}} = \tilde{A}_t^{\mathrm{TS}}\right)|\hat{p}_t, p_t, \theta\right]|\theta\right]$$
$$\leq \mathbb{E}\left[\mathbb{E}\left[(R^* - \theta^\top \tilde{A}_t^{\mathrm{TS}})|\hat{p}_t, p_t, \theta\right]|\theta\right]$$
$$= \mathbb{E}\left[\mathbb{E}\left[(R^* - \theta^\top A_t^{\mathrm{TS}})|\hat{p}_t, p_t, \theta\right]|\theta\right]$$
$$= \mathbb{E}\left[R^* - \theta^\top A_t^{\mathrm{TS}}|\theta\right],$$

where the inequality follows from the nonnegativity of $R^* - \theta^\top \tilde{A}_t^{\mathrm{TS}}$. On the second part of the sum, we have

$$\mathbb{E}\left[\mathbb{E}\left[(R^* - \theta^\top \tilde{A}_t^{\mathrm{ES}})\mathbb{I}\left(d_{\mathrm{KL}}(\hat{p}_t\|p_t) \leq \epsilon, \tilde{A}_t^{\mathrm{ES}} \neq \tilde{A}_t^{\mathrm{TS}}\right)|\hat{p}_t, p_t, \theta\right]|\theta\right]$$
$$\leq \mathbb{E}\left[\mathbb{E}\left[(R^* - R_*)\mathbb{I}\left(d_{\mathrm{KL}}(\hat{p}_t\|p_t) \leq \epsilon, \tilde{A}_t^{\mathrm{ES}} \neq \tilde{A}_t^{\mathrm{TS}}\right)|\hat{p}_t, p_t, \theta\right]|\theta\right]$$
$$= \Delta(\theta)\mathbb{E}\left[\mathbb{E}\left[\mathbb{I}\left(d_{\mathrm{KL}}(\hat{p}_t\|p_t) \leq \epsilon, \tilde{A}_t^{\mathrm{ES}} \neq \tilde{A}_t^{\mathrm{TS}}\right)|\hat{p}_t, p_t, \theta\right]|\theta\right]$$
$$\leq \sqrt{\epsilon/2}\,\Delta(\theta),$$

where the last inequality follows from the way we couple $\tilde{A}_t^{\mathrm{ES}}$ and $\tilde{A}_t^{\mathrm{ES}}$. Thus, the result follows. $\quad\square$

Combining Lemma 7 and Lemma 8 delivers a proof for our main result. In particular, we have

$$\mathrm{Regret}(T, \pi^{\mathrm{ES}}, \theta) = \sum_{t=0}^{T-1}\mathbb{E}\left[(\theta^\top A^* - \theta^\top A_t^{\mathrm{ES}})\mathbb{I}\left(d_{\mathrm{KL}}(\hat{p}_t\|p_t) > \epsilon^2/2\right)|\theta\right]$$
$$+ \sum_{t=0}^{T-1}\mathbb{E}\left[(\theta^\top A^* - \theta^\top A_t^{\mathrm{ES}})\mathbb{I}\left(d_{\mathrm{KL}}(\hat{p}_t\|p_t) \leq \epsilon^2/2\right)|\theta\right]$$
$$\leq \frac{1}{2}\epsilon\Delta(\theta)T + \mathrm{Regret}(T, \pi^{\mathrm{TS}}, \theta) + \frac{1}{2}\epsilon\Delta(\theta)T$$
$$= \mathrm{Regret}(T, \pi^{\mathrm{TS}}, \theta) + \epsilon\Delta(\theta)T,$$

where the inequality follows from Lemma 7, with $\delta = \epsilon/2$, and Lemma 8. $\qquad\square$