[Reviews · NeurIPS 2017]

Reviewer 1



This paper introduces ensemble sampling as an approach to approximate Thompson sampling. Ensemble sampling is useful for the cases where sampling from the posterior is intractable. The idea of ensemble sampling is to maintain and increamentally update an ensemble of models and sample uniformly from these models whenever an approximation to stochastic sampling is needed. Even though the whole idea is general, it is not formulated in general and is only built on two use cases, linear bandit and neural networks. Moreover, the analysis (convergence bounds) is only developed for the case of linear bandit and not for neural network. Overall, the paper is written and organized well; it is easy to follow and addresses an important problem. Below are some of my concerns. The procedures for updating models is not clearly motivated and explained. The authors suggest maintaining a covariance matrix and proposing the incremental updates in line 104 as one possible procedure but it is not clear how this selection is made and what are other possibilities. No motivation or justification is provided. Is there a science behind this that can guide us to design a new one? Two examples are introduced in the paper; one for linear bandit and the other for Neural Network. Analysis are not provided for the case of neural network even though that is the use case for which using ensemble sampling is helpful in practice. Note that stochastic sampling for linear bandit in tractable. Figure 1a shows that ensemble sampling is better in early iterations and worse in later iterations compared to Thompson sampling. Why is that? Moreover, Figure 1b is not consistent with figure 1a in this regard. Almost to t=100, ensemble sampling with any number of models (plotted in Figure 1a) is better than stochastic sampling.

Reviewer 2



This paper presents a particle filter like finite ensemble approximation version of the posterior (Thompson) sampling approach and theoretical analysis for the bandit problems with potentially complex models. The analysis establishes that the regret can be broken as the sum of two terms, the regret of the ideal Thompson sampling and the regret due to the error introduced by the finite filter. The proof is sound and makes use of tricks, such as the reduction of the union bound on the action counts independent of the action order. Experiments are conducted on Gaussian linear bandits and neural networks, and the results presented confirm the practicality of the proposed approach for mildly large ensembles. Generally, the work is of some practical use and significance, but the paper would be strengthened with more details for and more convincing experimental results, given the relatively light theoretical contribution.

Reviewer 3



This paper proposes ensemble sampling which essentially approximates Thompson Sampling for complex models in a tractable way. This will be useful in a variety of applications. There does not seem to be an obvious mistake in the proofs. However, there are a number of limitations of the proposed method. See below for detailed comments: 1. What is the underlying noise model for the rewards? Please clarify this. 2. The weights of the neural network are assumed to be Gaussian distributed. Why is this a valid assumption? Please explain. 3. Please explain under what assumptions can the proposed method be applied? Can it applied for other model classes such as generalized linear models, and SVMs? 4. A neural network offers more flexibility. But is this truly necessary for applications where the time horizon is not too large, and the number of samples are limited? 5. Missing references in lines 85 - 92. Please add citations wherever necessary. 6. In line 3 of Algorithm 1, the model is picked uniformly at random. Can the results be improved by using a more intelligent sampling scheme? 7. Using a sample from a randomly perturbed prior and measurements seems similar to the Perturb and MAP framework in [1]. Please cite this work and describe how your framework is different. 8. To update the models, it is necessary to store the entire history of observations. This is inefficient from both a memory and computational point of view. 9. In line 120, "some number of stochastic gradient descent iterations". Please clarify this. How many are necessary? Do you model the error which arises from not solving the minimization problem exactly? 10. In Theorem 3, the approximation error increases with T. The approximate regret is then in some sense linear in T. Please justify this. Also, what is the dimension dependence of the approximation error? Also, please give some intuition on the linear dependence of on the gap \delta. 11. In section 5, the experiments should show the benefit of using a more complex model class. This is missing. Please justify this. 12. A sampling based technique should be used as a baseline. This is also missing. Please justify. 13. In line 249, "we used a learning rate of.. ". How did you tune these hyper-parameters? How robust are your results to these choices? [1]. Gaussian Sampling by Local Perturbations,